# Effects of natural treatments on the varroa mite infestation levels and overall health of honey bee (*Apis mellifera*) colonies

Laura Narciso[1], Martina Topini[1,2☯], Sonia Ferraiuolo[1,2☯], Giovanni Ianiro[1], Cinzia Marianelli[1]*

1 Department of Food Safety, Nutrition and Veterinary Public Health, Istituto Superiore di Sanità, Rome, Italy,
2 Sapienza University of Rome, Rome, Italy

☯ These authors contributed equally to this work.
* cinzia.marianelli@iss.it

**Data Availability Statement:** All relevant data are within the manuscript and its Supporting Information files.

## Abstract

The survival of the honey bee (*Apis mellifera*), which has a crucial role in pollination and ecosystem maintenance, is threatened by many pathogens, including parasites, bacteria, fungi and viruses. The ectoparasite *Varroa destructor* is considered the major cause of the worldwide decline in honey bee colony health. Although several synthetic acaricides are available to control Varroa infestations, resistant mites and side effects on bees have been documented. The development of natural alternatives for mite control is therefore encouraged. The study aims at exploring the effects of cinnamon and oregano essential oils (EOs) and of a mixed fruit cocktail juice on mite infestation levels and bee colony health. A multi-method study including hive inspection, mite count, molecular detection of fungal, bacterial and viral pathogens, analysis of *defensin-1*, *hymenoptaecin* and *vitellogenin* immune gene expression, colony density and honey production data, was conducted in a 20-hive experimental apiary. The colonies were divided into five groups: four treatment groups and one control group. The treatment groups were fed on a sugar syrup supplemented with cinnamon EO, oregano EO, a 1:1 mixture of both EOs, or a juice cocktail. An unsupplemented syrup was, instead, used to feed the control group. While *V. destructor* affected all the colonies throughout the study, no differences in mite infestation levels, population density and honey yield were observed between treatment and control groups. An overexpression of *vitellogenin* was instead found in all EO-treated groups, even though a significant difference was only found in the group treated with the 1:1 EO mixture. Viral (DWV, CBPV and BQCV), fungal (*Nosema ceranae*) and bacterial (*Melissococcus plutonius*) pathogens from both symptomatic and asymptomatic colonies were detected.

## 1. Introduction

The value of honey bees (*Apis mellifera*) to the human population and to the whole ecosystem as pollinators of agricultural crops and wild plants worldwide, is undisputed. Unfortunately,

**Funding:** This work was supported by the Associazione Apicoltori Siena Grosseto Arezzo, Italy (https://www.asgamontalcino.com/) – grant 2022 to CM; the Istituto Superiore di Sanità, Italy. Both the funders were not involved in the study design, analysis and interpretation of data, the writing of this article or the decision to submit it for publication.

**Competing interests:** The authors have declared that no competing interests exist.

beekeepers have been reporting a general decline in honey bee populations, as well as an increase in colony losses especially in Western Europe (https://www.efsa.europa.eu/en/topics/insect-pollinator-health). The survival of honey bee population is threatened by numerous factors (often a combination of them), which include attacks of pathogens (parasites, bacteria, virus, fungi and protozoa) [1, 2], genetic factors, immunodeficiencies, loss of habitat, climate changes and the use of pesticides [3, 4]. The parasite *Varroa destructor* and the infections it carries (especially viruses) are one of the main causes of bee losses [5, 6].

A wide range of products for parasitosis control have been developed: synthetic compounds, also known as hard acaricides—e.g. flumethrin, formamidine amitraz and pyrethroids such as Tau-fluvalinate—and organic substances, also known as soft acaricides—e.g. formic acid, oxalic acid and essential oils (EOs) [7]. Although synthetic acaricides are easier to use and more effective against varroa mites than organic acaricides, their overuse has led to the emergence of resistance in treated parasites [8, 9], and to accumulation of the active ingredients and/or their metabolites in the bee products, with subsequent risk to the consumer [10]. Moreover, synthetic acaricides may be toxic for both larval and adult bees [11] or affect the physiology, metabolism and/or behaviour of honey bees [12]. Soft acaricides, such as EOs and organic acids, have aroused great interest over the years. They are eco-friendly (since they derive from plant species), usually safe for honey bees and beekeepers, and effective at lowering mite population levels [13].

EOs are highly volatile plant components with well-known properties, such as bactericidal, virucidal fungicidal and antiparasitic activities [14]. Terpenes, terpenoids and phenolic compounds are the major constituents of EOs and are the primary cause of their biological activity [14]. EOs and their monoterpenes are widely studied as alternatives to synthetic acaricides for controlling pests and diseases in honey bees [15–18]. However, few EO-based products have entered the market because of the discrepancy between independent field studies, even for the same plant species (14). The reason for this divergence has to be found in several factors, such as the variability of the chemical compositions of plants, experimental conditions of temperature and humidity, colony strength, degree of infestation/infection of the colony and experimental design, such as administration routes of EOs (e.g. vaporization, spraying, exposure to EO-embedded strips, EO-supplemented diets), doses and number of treatments [14, 19–23]. Differences are also documented between field and laboratory tests. Among the EOs found effective against varroa mites under controlled laboratory conditions, only a small number of EOs was effective when used directly in hives. The higher volatility of oils in open systems, the ventilation of bee workers within the hive, the different phytochemical profile of EOs tested in laboratory and field studies and environmental conditions, are some of the reasons which can explain the reduced *in vivo* effectiveness of EOs (14). Further investigations on EOs are therefore needed to establish effective field protocols, i.e. application method, dose, treatment repetition, time, etc.,—to prevent or reduce diseases in honey bee populations.

Fruit is an important food resource for humans and many animals, including insects, being a combination of vitamins, minerals, antioxidants, fibre and other phytochemicals. Honey bees have been observed foraging on fruit juice, depending on the context (fruit abundance and proximity, nectar dearth, etc.) and specific colony needs (e.g. water demand) [24]. In addition, bees fed with syrups enriched with fruit juices showed an increased productivity in comparison to control bees fed with regular sugar syrup [25]. Furthermore, dietary supplementations of aminoacids and vitamins to experimentally infected honey bees with *Nosema cerane* stimulated immune-related genes, increased the resistance to the infection and reduced bee mortality [26]. Dietary supplementation of wintering and early springing honey bees with vitamin C increased protein content and antioxidative enzyme activity in the freshly emerged worker bees, suggesting a potential enhancing health role of the vitamin [27].

However, to the best of our knowledge, the effects of fruit as natural sources of vitamins and antioxidants on bee health have not been thoroughly investigated.

The objective of the study was to explore the effects of natural treatments–i.e. cinnamon and oregano EOs and a juice cocktail–given to honey bees as dietary supplements to investigate possible new strategies for reducing varroa mite infestation and improving overall colony health. We chose to test cinnamon and oregano EOs as both oils had proven to be effective against Varroa mites and safe for adult honey bees under laboratory conditions [18]. As *V. destructor* feeds on the fat body of honey bees and negatively affects their immune responses, making them more susceptible to disease agents [28]; we tested the hypothesis that the varroacidal effect of EOs or the immunostimulatory properties of both EOs and fruit juices, or even the joint action of both varroicidal and immunostimulatory activities of EOs, counteract the mite infestation and, consequently, increase the bee health status. Specifically, we performed routine visual inspections of the colonies, measured the infestation levels of *V. destructor*, molecular detected pathogenic DNA (bacteria, fungi and protozoa) and RNA (virus) in the colonies and quantified the expression levels of three immune related genes (*defensin-1*, *hymenopataecin* and *vitellogenin*). We also looked into the adult bee population density and honey production throughout the study period in order to obtain in-depth knowledge of the effects of tested treatments on bee health.

## 2. Materials and methods

### 2.1 Field study: Experimental design and sample collection

The three-month trial is illustrated in **Fig 1**. Briefly, an experimental apiary of 20 hives of *Apis mellifera* subspecies *ligustica* was established in a flat area (42˚ 39′ 35.4″ N, 11˚ 5′ 53.8″ E) in the Maremma Regional Park, a natural park in the Region of Tuscany, Italy, from March 29th to June 28th, 2022.

Hives were randomly divided into five groups of four colonies (hives) each (four treatment groups and one control group): CIN (treated with the cinnamon EO), ORE (treated with the oregano EO), CIN+ORE (treated with a 1:1 mixture of cinnamon and oregano EOs), FRU (treated with a mixture of fruit juice with high vitamin and antioxidant ingredients), and UNT (untreated control group). All groups are shown in **Fig 1**. For further details, see the "Treatments protocols" section below.

Visual inspection of all colonies was conducted by experienced bee inspectors throughout the study to verify the effects of the experimental treatments on the colonies. Both bee brood and adult bee workers were routinely examined, and any abnormal behaviour or sign of illness was recorded. Adult bee population density and productivity were also documented throughout the 3-month trial (See section 2.7 below).

Samples of adult worker honeybees (about 30–50 bees) were collected in 50-mL tubes from both the control and treatment colonies at specific time points and whenever clinical signs or abnormalities (both in behaviour and brood pattern) were observed. Specifically, samples collected just before the administration of any treatment (T0), and 24 hours after the first (T1) and the third (T25) experimental treatments, as well as one month (T55) and two months (T85) after the last treatments, were used for the molecular diagnosis of both DNA and RNA pathogens (See section 2.5 below). Samples collected just before any treatment (T0), and 24 hours after the first (T1) and the third (T25) treatment, were instead used for the gene expression analysis (See section 2.6 below). The sampling is shown in **Fig 1** (green arrows).

All samples were immediately frozen in dry ice after collection and transported to Istituto Superiore di Sanità in Rome, where they were stored at -30˚C until used for DNA and RNA extractions (See section 2.4 below).

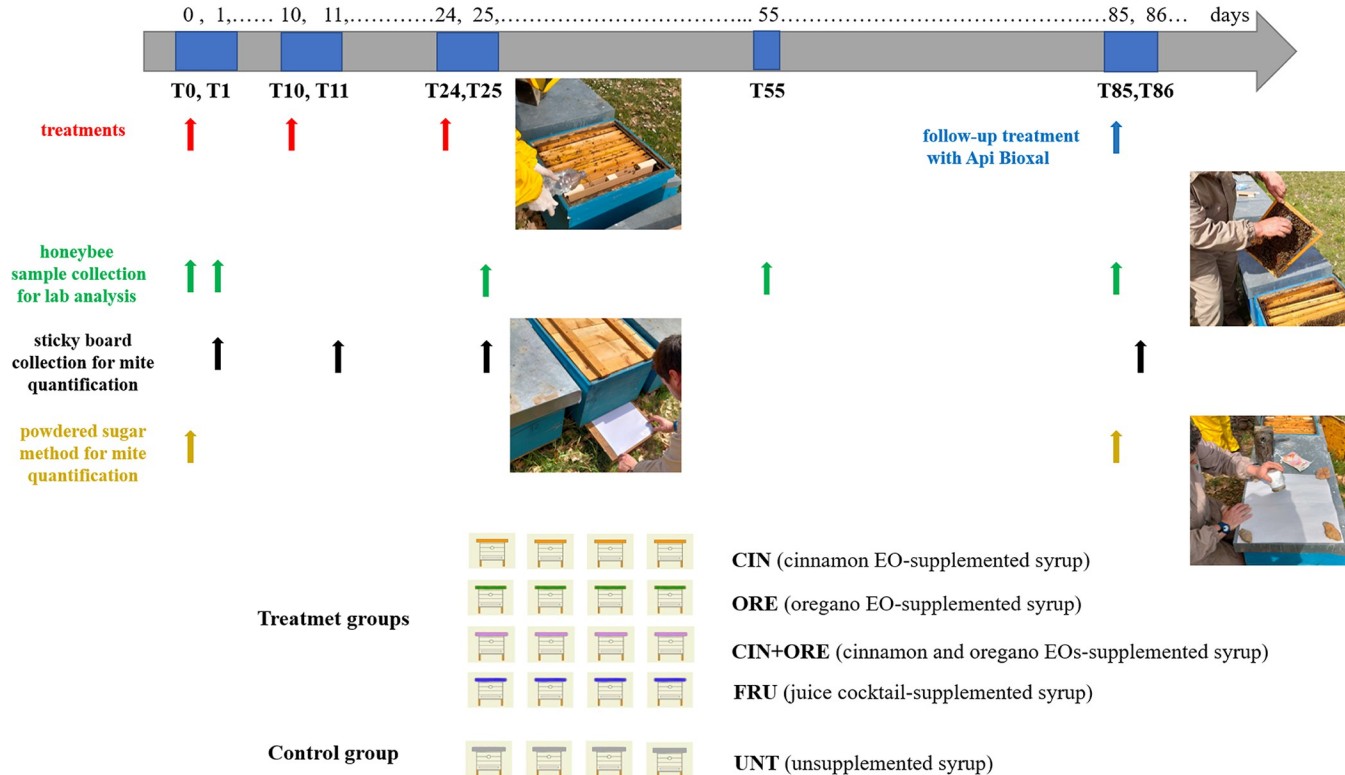

**Fig 1. Experimental design.** Twenty hives from the same apiary were divided into five homogenous groups, each of which was fed with a different kind of syrup for three times, approximately every 10–14 days: T0, T10 and T24 (three experimental treatments, red arrows). A final treatment with Api Bioxal was administered at the end of the study to all hives (blue arrow). Mite infestation levels were monitored over the study by two methods–the powdered sugar method (orange arrows) and the mite fall method (black arrows). Samples of adult bee workers were collected (green arrows) and were used for both the molecular diagnosis of DNA and RNA pathogens and the analysis of some immune-related genes in bees.

Changes in mite infestation levels were also monitored in both treatment and control groups by two methods–the powdered sugar and mite fall methods (See section 2.3 below).

## 2.2 Treatment protocols

A syrup solution containing 40% glucose and 30% fructose was prepared and supplemented with the cinnamon EO (*Cinnamomum zeylanicum*, Gisa Wellness, Italy), oregano EO (*Origanum vulgaris*, Gisa Wellness, Italy), a 1:1 mixture of both the EOs, or a juice cocktail. The cinnamon EO, oregano EO, cinnamon and oregano EO mixture and juice cocktail-supplemented syrups were used to feed the treatment groups CIN, ORE, CIN+ORE and FRU, respectively, as detailed below. An unsupplemented syrup was, instead, used to feed the control group (UNT). The groups and types of treatment are shown in **Fig 1**.

The cinnamon and oregano EO-supplemented syrups were prepared as follows. A phosphate-buffered saline (PBS) solution of soy lecithin (8 ng/mL) was prepared as per Zheng and colleagues [29] and used as emulsifier to stabilize the EO into the syrup. Specifically, 500 μL of cinnamon EO, oregano EO or the cinnamon EO+ oregano EO mixture (1:1, 250 μL+ 250 μL), were added to 10 mL soy lecithin solution, mixed and then pooled into 1 L syrup (0.05%). One litre EO-supplemented syrup/hive/treatment was used for the groups CIN, ORE and CIN +ORE.

The juice cocktail-supplemented syrup was prepared as follows. A fruit mixture (1:1) of a product containing 59% apple, 21% orange, 10% red turnip, 7% pomegranate, 2% lemon and

1% ginger juice (Cuore di Frutta, Only Juices srl, Milan, Italy; 0.49% protein, 10.75% carbohydrate, 0.42% fibre and 3% fat) and of a product containing 80% blueberry juice (A&D S.p.A. Gruppo Alimentare e Dietetico, Assago, Italy) was prepared and added to the syrup to make a 20% juice cocktail-supplemented syrup (200 mL fruit mixture were added to 1 L syrup). One litre of this syrup/hive/treatment was used for FRU.

The supplemented syrups were administered to the treatment groups, as well as the unsupplemented syrup to the control UNT group, three times–at intervals of about 10–14 days–by pooling the syrups (1L/hive) into the honeybee feeders: at T0, T10 and T24. The three experimental treatments are shown in **Fig 1** (red arrows).

A final treatment with Api Bioxal (Chemicals Laif SpA, Vigonza, Italy), a commercial oxalic acid- (acid oxalic dihidrat 62.0 mg/mL), glycerol-based product, was administered at the end of the study (T85) to all hives (both treatment and control groups) as follow-up treatment (**Fig 1**, blue arrow). Api Bioxal was used to kill and enumerate surviving mites, by dribbling 5mL per seam of bees, according to the manufacturer's instructions.

## 2.3 Quantification of mite infestation levels

The infestation levels of *V. destructor* mites were investigated throughout the study using the powdered sugar and mite fall methods. The powdered sugar was used to measure the differences in mite infestation levels between the beginning and the end of the study; the mite fall method was performed to evaluate the effectiveness 24 hours after each experimental treatment.

The powdered sugar method was performed on a sample of adult worker honeybees (about 50 g of bees), which was collected from each hive (both treatment and control colonies), at both T0 (just before any treatment) and T85 (the end of the study), as shown in **Fig 1** (orange arrows). Briefly, the collected bees were poured into a 750 mL jar provided with a mesh and a cap. Thirty-five grams of powdered sugar were added and the jar was closed immediately. Then, the jar was rotated for 3 minutes to cover all the bees with the sugar, and then inverted and shaked vigorously over a sheet of white paper allowing mites to drop off and fall through the mesh to the collection sheet. The mites obtained were then counted. Absolute changes in mite infestation levels were then calculated by subtracting the number of mites dropped from the bee workers at T0 from the number of mites at T85.

The mite fall method was carried out by placing sticky boards at the bottom of each hive (both treatment and control hives) just before the administration of each treatment. The sticky boards with the trapped mites were collected 24 hours after each experimental treatment–namely at T1, T11 and T25 –and 24 hours after the final treatment with Api Bioxal (T86), as shown in **Fig 1** (black arrows)–and transported to the laboratory for the mite-drop count. To estimate the number of varroa mites in a colony after each treatment and make the data comparable with each other, the number of trapped mites on each sticky board (n) was normalized to the size of the colony–calculated as frames of bees (f) in a 10-frame hive–and expressed with the formula

$$N = (n/f) \times 10$$

## 2.4 Sample preparation for DNA and RNA extractions

Twelve adult worker honeybees were randomly picked up from each sample and used for DNA and RNA extraction. The 12 bees were distributed in four GeneReady Grinding tubes (Life Real, China)–three bees per tube–with 800 μL of DNA/RNA Shield (Zymo Research,

USA). Tubes were then homogenised in the Ultracool GeneReady system (Life Real, China) by six shaking cycles at 6.50 m/s for 1 min at 4˚C, each followed by a rest time of 1 min. One hundred microliters of each of the four tubes per sample were pooled together (a total of 400 μL per sample) prior to DNA/RNA extraction. The Quick-DNA/RNA Miniprep Plus kit (Zymo Research, USA) was used for DNA/RNA isolation, following the manufacturer's instructions. The eluted DNA and RNA were stored at -30˚C until further analysis. The nucleic acid concentration and purity were measured by NanoDrop 1000 Spectrophotometer (Thermo Fisher Scientific Inc.). DNA samples were used for the detection of common DNA pathogens of bees [30]: *Melissococcus plutonius*, *Paenibacillus larvae*, *Nosema ceranae*, *Nosema apis* and *Malpighamoeba mellificae*. RNA samples, on the other hand, were used for both the detection of common RNA pathogens of bees [30]–deformed wing virus variants A (DWV-A) and B (DWV-B), acute bee paralysis virus (ABPV), chronic bee paralysis virus (CBPV), sacbrood bee virus (SBV) and black queen cell virus (BQCV)–and the study of honeybee immune response, as detailed below.

## 2.5 Molecular identification of DNA and RNA pathogens by PCR and nucleotide sequencing

The molecular diagnosis of both DNA and RNA pathogens circulating in the honey bee colonies was performed at T0 (before the administration of any treatment), and at T25, T55 and T85 (24 hours, one month and two months after the third and last experimental treatment, respectively).

Two μL of DNA from each sample were used as a template for the PCR assays to detect and identify DNA pathogens (Table 1). Reactions were performed in a total volume of 50 μL using MyTaq Red DNA Polymerase (Bioline, London, UK) following the manufacturer's instructions. The cycling conditions for all PCR amplifications were: denaturation at 95˚C for 2 minutes, followed by 35 cycles at 95˚C for 30 s, 58˚C for 30 s and 72˚C for 90 s, and a final extension cycle at 72˚C for 7 minutes. The reference strains *M. plutonius* CCM 3707 and *P. larvae* CCM 4484 purchased from the Czech Collection of Microorganisms (Masaryk University, Czech Republic) were used as controls.

To detect and identify RNA pathogens, up to 1 ng RNA from each sample was used as a template for reverse transcription. Reactions were performed in a total volume of 20 μL using SensiFAST cDNA Synthesis kit (Bioline, London, UK) following the manufacturer's

**Table 1. Primers used for PCR and sequencing of DNA pathogens.**

| DNA pathogen | Primer sequence (5'-3') (F = forward, R = reverse) | Size (pb) | Gene target | Reference |
|---|---|---|---|---|
| *Melissococcus plutonius* | F– GAA GAG GAG TTA AAA GGC GC | 831 | 16S rRNA | [31] |
| | R– TTA TCT CTA AGG CGT TCA AAG G | | | |
| *Paenibacillus larvae* | F– CTT GTG TTT CTT TCG GGA GAC GCC A | | 16S rRNA | [32] |
| | R– TCT TAG AGT GCC CAC CTC TGC G | 1106 | | |
| *Nosema ceranae* | F– CGG CGA CGA TGT GAT ATG AAA ATA TTAA | | 16S rRNA | [33] |
| | R– CCC GGT CAT TCT CAA ACA AAA AAC CG | 218 | | |
| *Nosema apis* | F– GGG GGC ATG TCT TTG ACG TAC TAT GTA | | 16S rRNA | [33] |
| | R– GGG GGG CGT TTA AAA TGG AAA CAA CTA TG | 321 | | |
| *Malpighamoeba mellificae* | F– TGT GTA AAA GCG ATT GGT AGA AAG | 600 | 16S rRNA | This study |
| | R– ACC CTA ATT GTT ACT CAC ATC GT | | ID OL757386.1* | |

* sequence ID from Genebank used as the input template for the Primer-BLAST tool at https://www.ncbi.nlm.nih.gov/tools/primer-blast/ for the design of target-specific primers.

**Table 2. Primers used for PCR and sequencing of RNA pathogens.**

| RNA pathogen | Primer sequence (5'-3') (F = forward, R = reverse) | Size(pb) | Gene target | Reference |
|---|---|---|---|---|
| DWV-A | F– GAT CGC TGA ACG TTG TAC GC | 372 | Viral polyprotein | This study |
| | R– GCC TGC ACC GGA TTC GAT AA | | ID OR361536.1* | |
| DWV-B | F– TAG GAA AAG ACG CGG GTG AG | 484 | Viral polyprotein | This study |
| | R– TCG GCA ATT TGA TAC CAA CGC | | ID OR361560.1 * | |
| ABPV | F– GGA TCC TGC CCC TTG TGA AT | 504 | Structural protein | This study |
| | R– GGT GGT TGA AAC ACT TGC GT | | ID AF486073.2* | |
| CBPV | F– GCT ACA TTT TGC GGT AGG CG | 353 | Polyprotein | This study |
| | R– TAG CTT CAG GTG GTA CGG A | | ID MF175173.1* | |
| SBV | F– ACG CTT AGC GCC TAT TTA CC | 397 | Structural protein | This study |
| | R– GAT TAG ACA CGT TTG CGG GC | | ID AF092924.1* | |
| BQCV | F– TAT TGA AGC ACC CCG TCT CG | 497 | Structural polyprotein | This study |
| | R– GAG CCG TCT GAG ATG CAT GA | | ID MT482476.1* | |

* sequence ID from Genebank used as the input template for the Primer-BLAST tool at https://www.ncbi.nlm.nih.gov/tools/primer-blast/ for the design of target-specific primers.

instructions. Four μL of cDNA from each reaction were then used as template for the PCR assays to detect RNA pathogens (**Table 2**). Reactions were performed in a total volume of 50 μL using MyTaq Red DNA Polymerase (Bioline, London, UK), as described above for the DNA pathogens.

All the PCR products were purified and then sequenced using the same PCR primers to confirm the results. Sequences were analysed using ABI Prism SeqScape software, version 2.0 (Applied Biosystems). The consensus sequences generated by matching forward and reverse reads and removing PCR primers were compared to sequence databases and identified using the BLAST tool (https://blast.ncbi.nlm.nih.gov/Blast.cgi).

The honeybee mitochondrial DNA (Genebank ID OM203347.1) was used to design a couple of primers (F - AAC ATC GAG GTC GCA AAC ATC and R– TTA GGT CGA TCT GCT CAA TGA A) for PCR amplification and sequencing of the partial 16S mitochondrial rRNA gene (a 426-bp fragment). Results confirmed the quality of the diagnostic process (from the nucleic acid extractions to PCR amplifications).

## 2.6 Gene expression quantification of honeybee immune system by using the reverse transcription quantitative real-time PCR (RT-qPCR)

The expression levels of the genes encoding for the antimicrobial peptides (AMPs) Defensin-1 and Hymenoptaecin, and for the glycolipoprotein Vitellogenin were studied at T0, T1 (before and 24 hours after the first experimental treatment) and T25 (24 hours after the third and last experimental treatment) in control and treatment colonies. Fold changes were used in the analysis of gene expression data as detailed below.

RNA extracts at T0, T1 and T25 were analysed to quantify relative gene expression of *defensin*-1, *hymenoptaecin* and *vitellogenin* by RT-qPCR (**Table 3**). cDNA was synthesised from RNA templates as described above and diluted 1:50 in molecular biology-grade water. Two microliters of this diluted cDNA were used for the qPCR. Reactions were performed using the SensiFAST SYBR Lo-Rox kit (Bioline, London, UK) according to the manufacturer's instructions and the following cycle condition: denaturation at 95°C for 2 minutes, followed by 40 cycles at 95°C for 5 s and 60°C for 30 s, and a final cycle at 95°C for 15 s, 65°C for 15 s and 95°C for 15 s for the melt-profile analysis. Quantitative PCR reactions were carried out on the

**Table 3. Primers used for qPCR.**

| Gene | Primer sequence (5'-3') (F = forward, R = reverse) | Size (pb) | Reference |
|---|---|---|---|
| *defensin*-1 | F– TGC GCT GCT AAC TGT CTC AG | 119 | [34] |
| | R – AAT GGC ACT TAA CCG AAA CG | | |
| *hymenoptaecin* | F– CTC TTC TGT GCC GTT GCA TA | 200 | [34] |
| | R– GCG TCT CCT GTC ATT CCA TT | | |
| *vitellogenin* | F– AGT TCC GAC CGA CGA CG | 62 | [35] |
| | R– TTC CCT CCC ACG GAG TCC | | |
| *β-actin* | F– TGC CAA CAC TGT CCT TTC TG | 155 | [36] |
| | R– AGA ATT GAC CCA CCA ATC CA | | |

AriaMx Real-Time PCR System (part number G8830A, Agilent Technologies, Santa Clara, CA, USA).

The expression levels of the target genes were normalized to that of the *β-actin* gene (ΔCt). Changes in the gene expression levels that took place 24 hours after the first treatment (ΔΔCt = ΔCt T1- ΔCt T0) and 24 hours after the third treatment (ΔΔCt = ΔCt T25 - ΔCt T0) were used to calculate the relative fold gene expression levels with the $2^{-\Delta\Delta Ct}$ formula.

## 2.7 Adult bee population density and honey production

Both adult bee population density and honey production were monitored over the study.

The adult bee population of each hive was estimated over the 3-month trial by enumerating the frames of bees, as previously described [37]. Absolute changes in colony density were then calculated by subtracting the number of bee frames at T0 from the number of bee frames at T55 (T55-T0) and T85 (T85-T0) in control and treatment colonies.

Each hive was provided with one or two 9-frame honey suppers, according to needs. As a full honey super could store about 15 kg of honey for an average 9-frame setup, while a frame held around 1.5 kgs, we estimated the quantity (kg) of honey produced by the number of frames filled with honey: e.g. zero kg were assigned when both the honey suppers were empty, 13.5 kgs when the first 9-frame supper was completely full of honey, and 27 kgs when also the second 9-frame supper was filled with honey. Honey production was measured at T55 and T85 in control and treatment colonies.

## 2.8 Statistical analysis

The Shapiro-Wilk test was used to assess data distribution. Median values per group and inter-quartile ranges were used to graphically describe changes in mite infestation levels and immune gene expression in both treatment and control groups by box plots. In particular, the minimum value (line at the bottom of the lower whisker), the 1st quartile (bottom of the box), the median (line inside the box), the 3rd quartile (top of the box), and the maximum value (line at the top of the upper whisker) are reported.

Absolute changes in mite infestation levels obtained by using the powdered sugar method– calculated by subtracting the initial value at T0 from the final value at T85 –were considered for the nonparametric statistical analysis. Between-group analyses (treatment groups vs. control) were performed by means of the Kruskal-Wallis test followed by Dunn's multiple comparisons test where applicable.

The normalized quantity of mites per colony obtained by the mite fall method 24 hours after each experimental treatment (T1, T11, T25) and 24 hours after the final treatment with Api Bioxal (T86), were considered for the nonparametric statistical analysis. Intra-group

differences of data sets recorded at T1, T11 and T25 were determined by the Friedman test, followed by Dunn's pairwise multiple comparisons test (T1 *vs* T11, T1 *vs* T25 and T11 *vs* T25). Inter-group analyses of differences between the control group and treatment groups at different time points (T1, T11, T25 and T86) were performed by Kruskal-Wallis test followed by the Dunn's multiple comparisons test where applicable.

Relative fold gene expression levels of *defensin-1*, *hymenoptaecin* and *vitellogenin* at T1 and T25 were considered for the nonparametric statistical analysis. Between-group analyses (treatment groups *vs* control) were performed by the Kruskal-Wallis test followed by the Dunn's multiple comparisons test where applicable.

Absolute changes in adult bee population density at T55 and T85 –calculated by subtracting the number of bee frames at T0 from the number of bee frames at T55 and T85 –and honey production at T55 and T85 were considered for the nonparametric statistical analysis. Between-group analyses (treatment groups vs. control) were performed by the Kruskal-Wallis test followed by the Dunn's multiple comparisons test where applicable.

GraphPad Prism 9 version 9.0.0 for Windows, GraphPad Software, La Jolla California USA, www.graphpad.com, was used for all statistical analyses. Differences were considered significant at $P < 0.05$.

## 3. Results

### 3.1 Safety of treatments

The treated bees seemed to like the supplemented syrups as much as the bees of the control UNT group liked the unsupplemented syrup. Both supplemented and unsupplemented syrups were consumed and stored completely within a few hours after the administration.

### 3.2 Infestation levels of *V. destructor*

Results are shown in **Fig 2** and S1 Fig.

The *V. destructor* mite affected all colonies–throughout the study. The mite infestation levels (T85-T0) obtained by counting the sugar-dropped mites from adult bee workers, remained almost unchanged between the treatment groups (CIN, ORE, CIN+ORE and FRU) and the control group (UNT) and no significant differences were found (P = 0.273) (**Fig 2**, Panel A).

The amount of mite fallen 24 hours after each experimental treatment (T1, T11 and T25) on sticky boards was similar in all treatment groups to the natural mite falls recorded in the control group at the same times (**Fig 2**, Panel B). A significant intra-group decline in the mite fall was only found in FRU by comparing T25 *vs* T11 (P = 0.04; **Fig 2**, Panel B).

At the end of the study (T86) the final treatment with Api Bioxal, which was administered to all the hives, showed no significant differences in fallen mite levels between groups (**S1 Fig**).

### 3.3 Clinical signs and molecular diagnosis of both DNA and RNA pathogens

Results of clinical investigations and molecular detection of bee pathogens at T0, T25, T55 and T85 are shown in **Table 4**.

At T0, some bees of CIN suffered from bee tremble and flight inability (hives n. 1 and 4), compatible with CBPV infection. Molecular investigations, however, did not show the presence of the virus in the bee workers randomly collected from these colonies. On the other hand, CBPV was detected in adult bee samples from four asymptomatic colonies (hives n. 5, 14, 17 and 19). Ten days later, a large amount of dead bees was found in front of the hives n. 1 and 4, and in additional groups: ORE (hive n. 6), CIN+ORE (hive n. 11), FRU (hives n. 13 and

A    Absolute changes in mite infestation levels (T85-T0) by the powdered sugar method

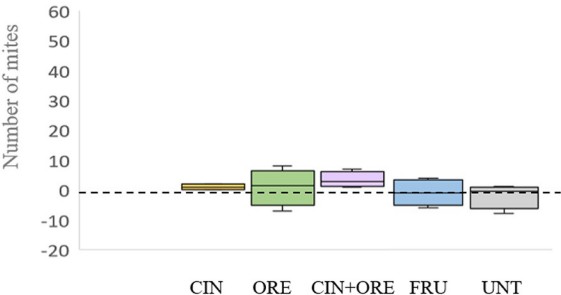

B    Mite quantity at T1, T11 and T25 by the mite fall method

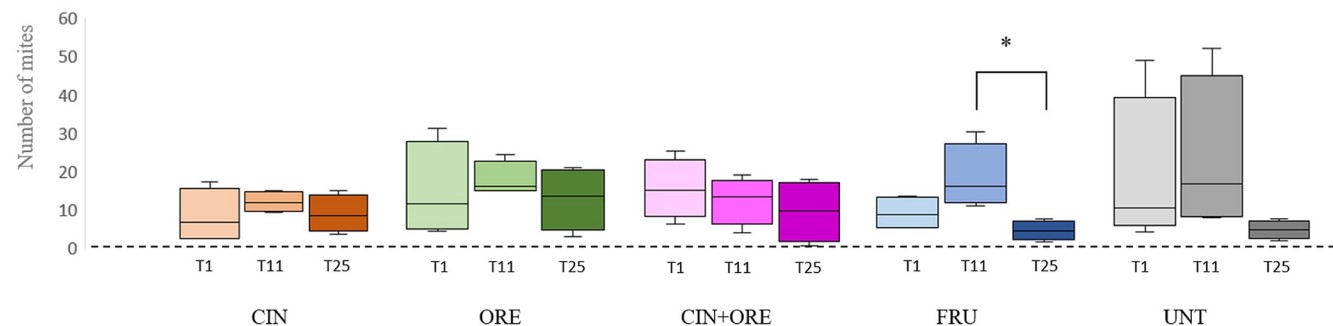

**Fig 2. Box plots of the mite infestation data.** The Fig shows the median absolute differences (Panel A) in mite levels and normalized median mite quantities (Panel B) and interquartile ranges in all treatment and control groups. The groups are: CIN, cinnamon EO-supplemented syrup; ORE, oregano EO-supplemented syrup; CIN+ORE, cinnamon and oregano EOs-supplemented syrup; FRU, juice cocktail-supplemented syrup; UNT, unsupplemented syrup. Between-group analysis by the Kruskal-Wallis test followed by the Dunn's multiple comparisons test was performed to determine the effects of the treatments on mite infestation levels. Intra-group differences of data sets recorded at T1, T11 and T25 were determined by the Friedman test, followed by Dunn's pairwise multiple comparisons test. * $P \le 0.05$. Data including minimum score, 1st quartile, median, 3rd quartile and maximum score are shown for each group.

14) and UNT (hives n. 18 and 20). The CBPV pathogen was detected in all the dead bee samples collected in front of the entrance of those hives. Either DWV-A or DWV-B was detected in a large number of colonies, even in the absence of clinical evidence: groups CIN (hive n. 4), FRU (hives 13, 14 and 15) and UNT (hives n. 19 and 20). The fungal *N. ceranae* and the bacterial *M. plutonius* pathogens were detected in adult bee samples from asymptomatic colonies (hives n. 13 and 20, respectively).

At T25, no clinical picture of CBPV infection was recognized in the apiary. A spotty brood appearance was the only clinical sign found, which affected one colony (hive n. 13, FRU). All colonies, with the exception of hive n. 20 (UNT), tested positive for BQCV (95%). DWV-B was detected in hive n. 13 (FRU), while both DWV variants were found in hive n. 20 (UNT). DNA pathogens were found in four healthy colonies: *N. ceranae* in hive n. 9 (CIN+ORE), *M. plutonius* in hives n. 17 and 20 (UNT), and both *N. ceranae* and *M. plutonius* in the hive n. 19 (UNT).

At T55, the clinical picture of hive n. 13 (FRU) got worse: spotty brood patter with small, yellow dying larvae in some cells were found. *M. plutonius* was detected in both yellow larvae and adult bee workers from that colony. The DNA pathogens *N. cerasae* and *M. plutonius* were found out in the healthy adult bee workers of hives n. 4 (CIN) and n. 19 (UNT),

**Table 4. Results of the molecular diagnosis of DNA and RNA pathogens.**

| Colony number | Group | T0 | | | T25 | | | T55 | | | T85 | | |
|---|---|---|---|---|---|---|---|---|---|---|---|---|---|
| | | Visual inspection | DNA pathogen | RNA pathogen | Visual inspection | DNA pathogen | RNA pathogen | Visual inspection | DNA pathogen | RNA pathogen | Visual inspection | DNA pathogen | RNA pathogen |
| 1 | CIN | bee tremble and flight inability | | | | | BQCV | | | BQCV | | | BQCV |
| 2 | CIN | | | | | | BQCV | | | | | | BQCV |
| 3 | CIN | | | | | | BQCV | | | BQCV | | | BQCV, DWV-B |
| 4 | CIN | bee tremble and flight inability | | DWV-A | | | BQCV | | *N. ceranae* | BQCV | | | BQCV, DWV-B |
| 5 | ORE | | | CBPV | | | BQCV | | | BQCV | | | BQCV |
| 6 | ORE | | | | | | BQCV | | | | | | BQCV |
| 7 | ORE | | | | | | BQCV | | | BQCV | | | BQCV |
| 8 | ORE | | | | | | BQCV | | | BQCV | | | |
| 9 | CIN +ORE | | | | | *N. ceranae* | BQCV | | | | | | |
| 10 | CIN +ORE | | | | | | BQCV | | | BQCV | | | BQCV |
| 11 | CIN +ORE | | | | | | BQCV | | | BQCV | | | BQCV |
| 12 | CIN +ORE | | | | | | BQCV | | | BQCV, DWV-B | | | BQCV |
| 13 | FRU | | *N. ceranae* | DWV-B | spotty brood pattern | | BQCV, DWV-B | spotty brood pattern with dying yellow larvae | *M. plutonius** | BQCV | Numerous dead and dying yellow larvae | *M. plutonius* ° | BQCV |
| 14 | FRU | | | CBPV, DWV-A | | | BQCV | | | BQCV | | | BQCV |
| 15 | FRU | | | DWV-A | | | BQCV | | | BQCV, DWV-B | | | BQCV |
| 16 | FRU | | | | | | BQCV | | | BQCV | | | BQCV |
| 17 | UNT | | | CBPV | | *M. plutonius* | BQCV | | | BQCV | | | BQCV |
| 18 | UNT | | | | | | BQCV | | | BQCV | | | BQCV |
| 19 | UNT | | | CBPV, DWV-B | | *M. plutonius, N. ceranae* | BQCV | | *M. plutonius* | BQCV, DWV-B | | | BQCV |
| 20 | UNT | | *M. plutonius* | DWV-A | | *M. plutonius* | DWV-A, DWV-B | | | BQCV | | | BQCV, DWV-A, DWV-B |

DNA pathogens investigated: *M. plutonius*, *P. larvae*, *N. ceranae*, *N. apis* and *M. mellificae*: RNA pathogens investigated: DWV-A (deformed wing virus), DWV-B (a DWV-A variant), ABPV (acute bee paralysis virus), CBPV (chronic bee paralysis virus), SBV (sacbrood bee virus) and BQCV (black queen cell virus).

\* *M. plutonius* was detected in both bee larvae and adult worker samples.

° *M. plutonius* was only detected in the bee larvae sample.

respectively. BQCV was still the prevalent virus detected in the apiary, affecting almost the totality of the colonies (85%). DWV-B was found in three colonies (hives n. 12, 15 and 19).

At T85, *M. plutonius* infection of hive n. 13 got even worse by affecting numerous larvae. DNA of the pathogen was only found in the larvae sample of the hive. No additional DNA pathogens were detected in the apiary. BQCV was still the prevalent virus detected at T85 (90%). DWV-B was found in the apiary in hives n. 3 and 4, and together with the other variant (co-infection DWV-A and DWV-B) again in hive n. 20.

## 3.4 Expression levels of immune related genes

Differences in the expression of the immune-related genes encoding for the AMPs Defensin-1 and Hymenoptaecin, and the glycolipoprotein Vitellogenin at T1 and at T25 are shown in **Fig 3**.

Both at T1 and T25, the expression levels of *defensin-1* and *hymenoptaecin* remained almost unchanged between the treatment groups (CIN, ORE, CIN+ORE and FRU) and the control (UNT) (**Fig 3,** panels A and B). On the other hand, the expression levels of *vitellogenin* appeared to increase in all EOs-treated groups (CIN, ORE and CIN+ORE), where a high intra-group variability was recorded (see the interquartile range of the data in **Fig 3**, panel C). However, significant differences were only found between CIN+ORE *vs* UNT. We detected a 1500-fold increase at T1 (P = 0.0062) and a 100-fold increase at T25 (P = 0.0136) in the *vitello-genin* gene expression in comparison with UNT (**Fig 3**, panel C).

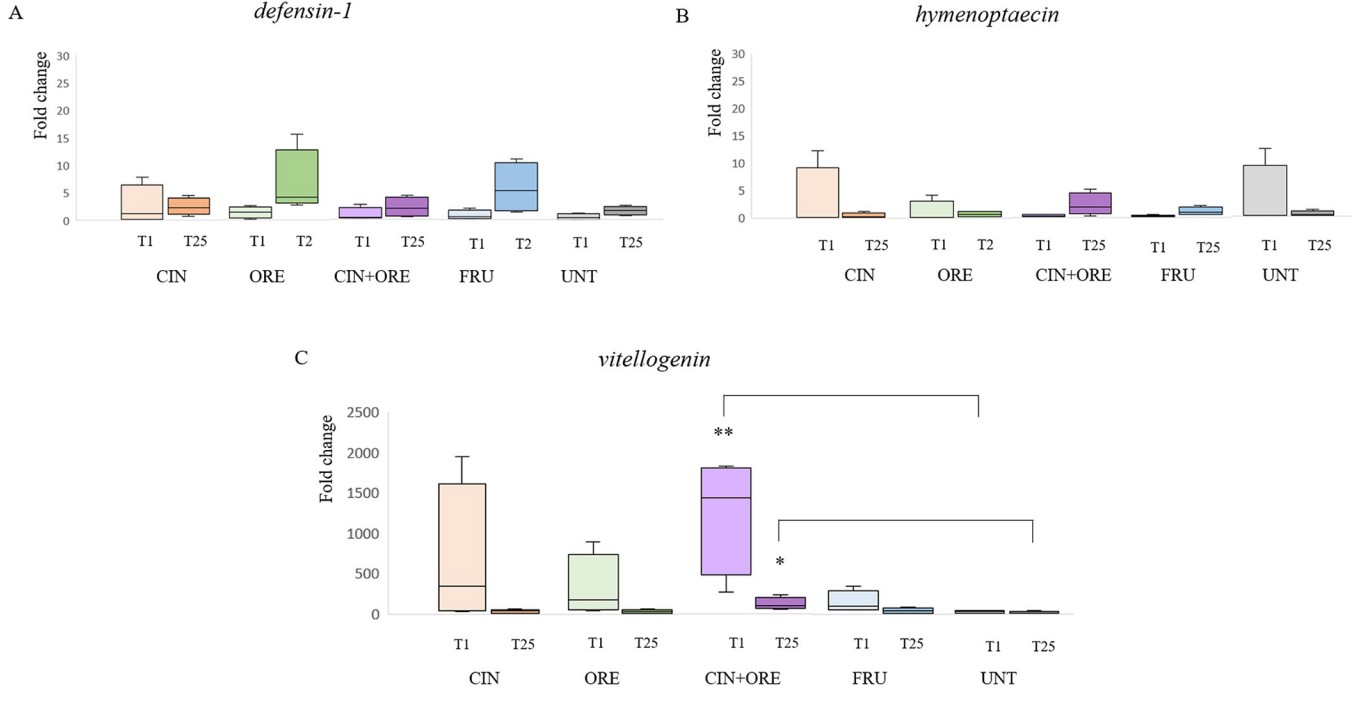

**Fig 3. Box plots of changes in gene expression at T1 and T25.** The Fig shows the fold change in expression of the genes *defensin-1* (Panel A), *hymenoptaecin* (Panel B) and *vitellogenin* (Panel C) in adult bee workers 24 hours after the first treatment (T1). The groups are: CIN, cinnamon EO-supplemented syrup; ORE, oregano EO-supplemented syrup; CIN+ORE, cinnamon and oregano EOs-supplemented syrup; FRU, juice cocktail-supplemented syrup; UNT, unsupplemented syrup. Between-group analysis by the Kruskal-Wallis test followed by the Dunn's multiple comparisons test was performed to determine the effects of treatments on gene expression levels. * P ≤ 0.05 and ** P ≤ 0.01. Data including minimum score, 1st quartile, median, 3rd quartile and maximum score are shown for each group.

### 3.5 Changes in adult bee population density and honey production

Changes in colony density at T55 and T85 in both control and treatment groups are shown in **Fig 4**.

An overall increase in colony density was observed in bees at both T55 and T85 in all hives (**Fig 4**, panels A and B, respectively). This increase was more substantial at T85. No significant differences were found between treatment and control groups.

In our field study, all colonies were able to produce honey with the exception of hive n. 13. The amount of honey in the honey suppers placed on the top of each hive was measured one month (T55) and two months (T85) after the third and last treatment (**Fig 5**). At T55, the median amount of honey was above one half-depth supper in all groups (**Fig 5**, panel A). At T85, a further month later, the median honey production ranged between one and two full-depth suppers in all groups (**Fig 5**, panel B). No significant differences were found between treatment and control groups.

## 4. Discussion

This field study explores the effects of the oral administration of natural treatments–i.e. cinnamon and oregano EOs and a juice cocktail–on *V. destructor* infestation levels and honeybee colony health.

The safety of EOs on *A. mellifera* is documented in the literature [38–40]. However, we did not investigate the toxicity of our treatments on bees. *V. destructor* was detected in all hives throughout the study. The oregano EO, cinnamon EO and the mixture of the two did not show any significant effect on the mites until then. The small group size of our study may have limited our findings. Many field studies have shown that several EOs and their main components (such as thymol and carvacrol) are effective in controlling Varroa mites under laboratory conditions [40–42]. Oregano and cinnamon EOs showed a high level of mite toxicity and a low level of bee toxicity *in vitro* [18]. However, the effects of EOs on mite control are often discordant between laboratory and field studies [37] and even among field trials [22, 43]. Many factors may explain the discrepancy in the results, such as the delivery method, the concentration or the chemical compositions of EOs used, season, temperature, bees and brood population, etc. EOs have been shown to release differently depending on the support, which affected the volatilization rate [23]. The continuous release of oregano EO by a vaporizer has been proven to have a high varroicidal efficacy in hives [22]. All the above suggests that the delivery method and dose of EOs may substantially affect the mite control results.

The mite *V. destructor* is considered a viral reservoir and a vector of some honeybee-associated virus, such as DWV and ABPV [4]. We investigated the two most commonly detected DWV variants, i.e. DWV-A and DWV-B. Although *V. destructor* infested all hives at T0 and remained substantially unchanged over the study despite the treatments, we detected the DWV RNA of both variants in very few colonies. One DWV-A/DWV-B co-infection was found, even if a potential helper effect between co-infecting variants has been documented [44].

The clinical sign of trembling was detected in a couple of bee colonies at T0. We were unable to detect the viral RNA in the randomly collected samples of bee workers from the two hives at that time. It is conceivable that very few or none of the affected bees were picked up for the RNA extraction and subsequent analysis. At T0, we instead detected CBPV in four asymptomatic colonies. In a previous study no CBPV positive sample was found in healthy bees [45]. The clinical signs of CBPV infection we observed in the apiary–i.e. trembling and death bees in front of the entrance resolved spontaneously in a few days regardless of whether they were observed in treated or control groups. Even if we did not investigate the CBPV titres

A

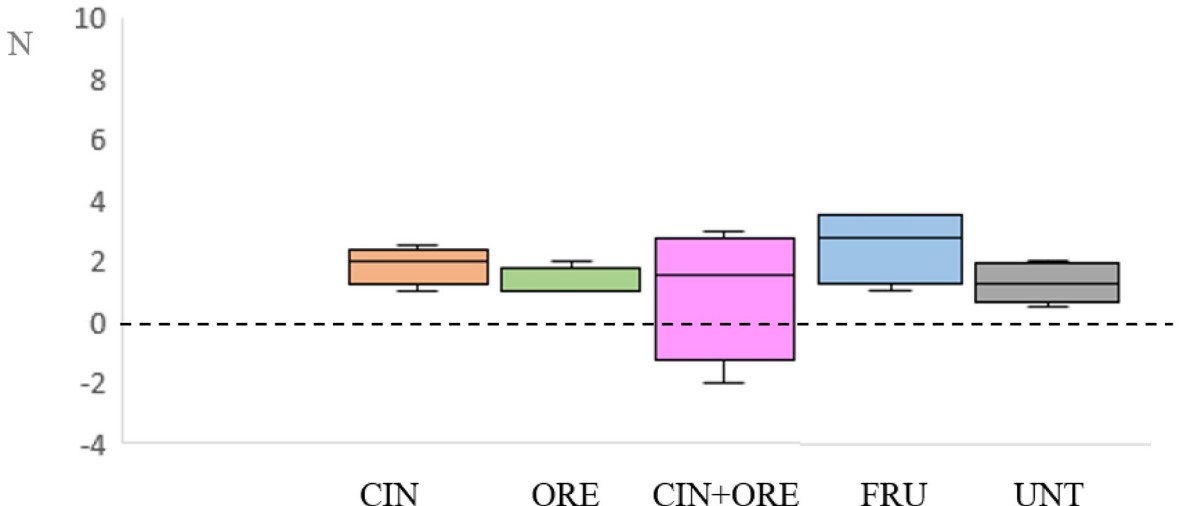

B

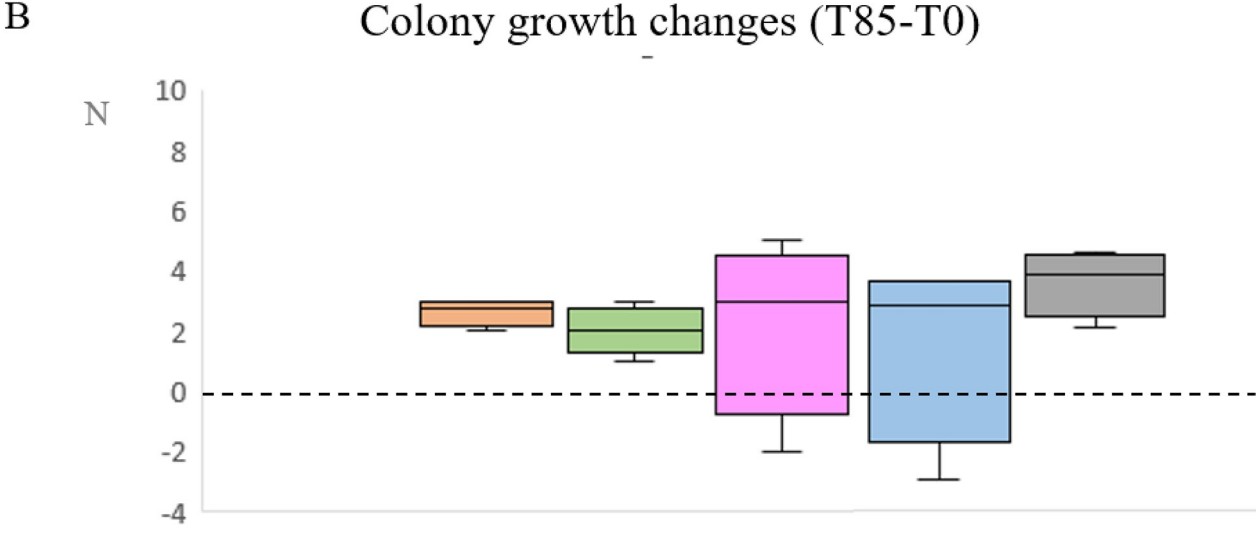

**Fig 4.** Box plots of changes in adult bee population density at T55 (panel A) and T85 (panel B). The adult bee population of each hive was estimated by enumerating the frames of bees. Differences in the density (N) were then calculated by subtracting the number of bee frames at T0 from that at T55 (Panel A) or T85 (Panel B). The dotted line represents the baseline. The groups are: CIN, cinnamon EO-supplemented syrup; ORE, oregano EO-supplemented syrup; CIN+ORE, cinnamon and oregano EOs-supplemented syrup; FRU, juice cocktail-supplemented syrup; UNT, unsupplemented syrup. Between-group analysis by the Kruskal-Wallis test followed by the Dunn's multiple comparisons test was performed to determine the effects of treatments on colony density. Data including minimum score, 1st quartile, median, 3rd quartile and maximum score are shown for each group.

A

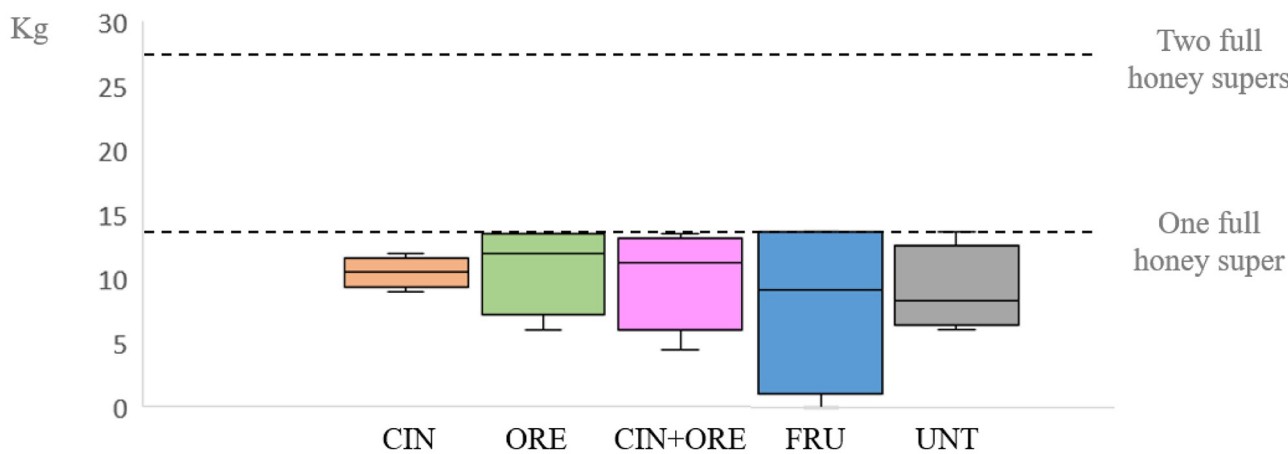

B

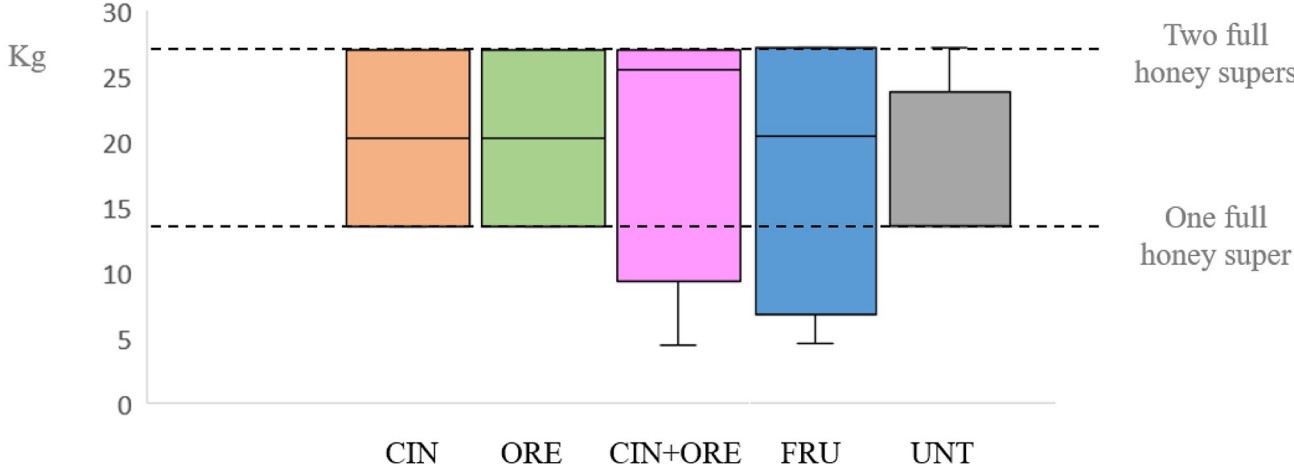

**Fig 5.** Box plots of honey production at T55 (panel A) and T85 (panel B). Honey production was estimated at T55 (Panel A) or T85 (Panel B) by the number of frames filled with honey. The groups are: CIN, cinnamon EO-supplemented syrup; ORE, oregano EO-supplemented syrup; CIN+ORE, cinnamon and oregano EOs-supplemented syrup; FRU, juice cocktail-supplemented syrup; UNT, unsupplemented syrup. Between-group analysis by the Kruskal-Wallis test followed by the Dunn's multiple comparisons test was performed to determine the effects of treatments on honey yield. Data including minimum score, 1$^{st}$ quartile, median, 3$^{rd}$ quartile and maximum score are shown for each group.

of the colonies, this finding seems to suggest that, whatever the viral loads were in our apiary, both asymptomatic and affected bees were able to overcome the CBPV infection.

We detected BQCV from almost all the colonies at T25 and at the subsequent time points T55 and T85. No colonies showed the clinical signs of BQCV infection, even if the prevalence ranged from 85% to 95% throughout the study. Our findings are in agreement with previous studies where high BQCV prevalence rates, such as 86% [46], 89.43% [47] and from 96.30 to

100% [45], were documented also in healthy honeybees [45, 48]. This data supports the previous observation that BQCV is less pathogenic than other bee viruses and, consequently, often found in clinically healthy colonies [45, 46].

*N. ceranae* and *M. plutonius* were the only DNA pathogens found in our apiary throughout the study. *N. ceranae* have been reported to parasitize on adult honeybees and lead to the death of diseased colonies more often than *N. apis* [4, 33]. During our study, *N. ceranae* was detected in a few sporadic colonies, which were all asymptomatic. It is likely that the infection level of the fungal pathogen was very low in our apiary. On the other hand, *M. plutonius* was molecular diagnosed from both asymptomatic and affected colonies. Specifically, our molecular tests revealed the presence of *M. plutonius* in samples of adult bee workers from both asymptomatic and symptomatic colonies, as well as from clinically affected larvae. Our findings seem to suggest that adult bee workers are suitable candidates for *M. plutonius* monitoring in the bee colony. Adult bee workers have been indeed found to better reveal the presence of *M. plutonius* than brood, even in the absence of clinical signs of disease [49], and are, therefore, considered more appropriate than brood for epidemiological studies of the disease [50]. The detection of the pathogen from asymptomatic colonies in our study, is concordant with the results of previous researches [51–53] and supports the view that bees can survive and proliferate even carrying the *M. plutonius* pathogen. It is conceivable that the risk for asymptomatic bees to become symptomatic increases when the bacterial titre rises or the bee immune system is somehow impaired (e.g. by other infectious agents or after exposure to pesticides). We could not estimate the effects of the treatments against either the viral, fungal or bacterial infections found in the apiary because the results were not conclusive due to the limited number of data or the lack of differences between the control and the treatment groups.

For a complete and clear picture of the effects of the natural treatments on bee health, we also looked into the colony density and honey production. The treatments did not affect either the colony density or honey production. This result did not surprise us, since honey yield depends mostly on the number of foraging bees. Several investigations have indeed proved positive correlation between stored pollen, brood production, honey yield and population size [54–56]. It has been observed that the addition of same EOs—such as cloves, mint, cinnamon, oregano, thyme, rosemary and basil -to the sugar syrup given to the bees, reduced the total number of germs in the intestine of bee workers and increased the capability of bees to collect nectar [57]. Moreover, those EOs increased brood and honey production, which was correlated with the development of the colonies [58]. Furthermore, the honey showed enhanced antioxidant and antimicrobial activity [59].

AMPs are recognized as key components of humoral innate immunity in honeybees [60]. AMPs—e.g. Apidaecins, Abaecin, Defensin 1, Defensin 2 and Hymenoptaecin–are secreted by the bee fat body in response to microbial infections or septic wounding [34]. A recent study has suggested that the understanding of gene expression of some of these immune genes, may serve as biomarkers for monitoring bee health status [47]. In this study, we have investigated the effects of the natural treatments on the expression levels of the immune genes *defensin-1* (encoding for the AMP Defensin 1), *hymenoptaecin* (encoding for the AMP Hymenoptaecin) and *vitellogenin* (encoding for the glycoprotein Vitellogenin), in order to better describe the health picture of the bee colonies. We found that 24 hours after the first (T1) and the third (T25) treatments, the gene expression levels of both *defensin-1* and *hymenoptaecin* were not statistically different between the groups.

The gene expression of *vitellogenin*, on the other hand, appeared to be up-regulated in all EOs-treated groups, even though significant differences were only found in CIN+ORE at both investigated time points: the *vitellogenin* expression was about 1500-fold up-regulated at T1 and 10-fold up-regulated at T25. It is conceivable that a larger group size may reveal

statistically significant relationships also in the other treatment groups. The small group size (four hives per group) and the high intra-group variability we recorded, might have limited the statistical power of our analysis. Our results are in contrast with a previous study where the expression levels of *vitellogenin* were reduced in bees fed on carvacrol- or thymol-enriched diets [61]. Vitellogenin is a multifunctional protein synthetized in the bee fat body, like AMPs. Vitellogenin has recognized nutritional and immunological functions. It is an egg-yolk precursor protein, that promotes bee longevity, binds to and eliminates bacterial and fungal pathogens, protects host cells from oxidative stress, binds to damaged host cells and protects them from further injury and transports zinc to maintain innate immunity [62, 63]. Vitellogenin is also involved in immune priming, namely it acts as a transporter of bacterial and viral fragments (immune elicitors) to offspring [62, 63]. Increased *vitellogenin* gene expression has been linked to decreased viral titres in bees [64]. We do not know the effects produced by the up-regulation of *vitellogenin* in CIN+ORE, since we did not investigate the viral loads. We did not find, however, any association between the *vitellogenin* expression levels and susceptibility to viral infection. Because of the important role played by Vitellogenin as an immunological and nutritional protein, further studies are needed to shed light on the effects that changing *vitellogenin* expression levels may have on bee health.

In conclusion, our study aimed at contributing to the development of effective, natural and safe options for the prevention/treatment of bee diseases by a multimethod approach. Although the tested natural products did not exhibit any effect against the mite *V. destructor* under our experimental conditions, we found all treatments to be safe for honeybees since they did not affect either the adult bee density or honey production. In addition, the mixture of both the CIN and ORE EOs exhibited the ability to up-regulate the gene expression of *vitellogenin*. As we do not know the consequences of the *vitellogenin* enhancement, additional investigations are required given the important role the protein Vitellogenin plays. Notwithstanding the limitations and challenges of applying EOs to bees under field conditions, EOs are promising options in mite control and colony health improvement. Further studies are therefore needed to investigate the most effective application methods and concentrations of the EOs at the colony level before defining standard operating procedures for beekeepers. The multimethod research approach here described may be used for an in-depth assessment of treatment efficacy and safety in apiaries.

## Supporting information

**S1 Fig. Box plot of mite infestation levels after the Api Bioxal treatment.** The Fig shows the normalized median mite quantities and interquartile ranges in both control and treatment groups. Number of mites trapped on sticky boards 24 hours after the treatment with Api Bioxal (T86). The groups are: CIN, cinnamon EO-supplemented syrup; ORE, oregano EO-supplemented syrup; CIN+ORE, cinnamon and oregano EOs-supplemented syrup; FRU, juice cocktail-supplemented syrup; UNT, unsupplemented syrup. Between-group analysis was performed by the Kruskal-Wallis test followed by the Dunn's multiple comparisons test. Data including minimum score, 1st quartile, median, 3rd quartile and maximum score are shown for each group.
(TIF)

## Acknowledgments

We gratefully thank the beekeepers Massimo Ciabini, Luca Rampini and Lorenzo Rossetti for their skilled technical assistance in this study. We also thank Dr. Flavia Chiarotti of Istituto Superiore di Sanità for her critical revision of the statistical analysis.

## Author Contributions

**Conceptualization:** Laura Narciso, Cinzia Marianelli.

**Formal analysis:** Cinzia Marianelli.

**Funding acquisition:** Cinzia Marianelli.

**Investigation:** Laura Narciso, Martina Topini, Sonia Ferraiuolo, Giovanni Ianiro, Cinzia Marianelli.

**Methodology:** Laura Narciso, Cinzia Marianelli.

**Project administration:** Cinzia Marianelli.

**Supervision:** Cinzia Marianelli.

**Visualization:** Cinzia Marianelli.

**Writing – original draft:** Cinzia Marianelli.

**Writing – review & editing:** Laura Narciso, Martina Topini, Sonia Ferraiuolo, Giovanni Ianiro, Cinzia Marianelli.

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
