## [Decision Letter · Decision Letter 0]

15 Jan 2024

PONE-D-23-42744A multi-method field study to investigate the effects of natural treatments on the health of honey bee (Apis mellifera) coloniesPLOS ONE

Dear Dr. Marianelli,

Thank you for submitting your manuscript to PLOS ONE. After careful consideration, we feel that it has merit but does not fully meet PLOS ONE’s publication criteria as it currently stands. Therefore, we invite you to submit a revised version of the manuscript that addresses the points raised during the review process.

We look forward to receiving your revised manuscript.

Kind regards,

Kai Wang

Academic Editor

PLOS ONE

Journal Requirements:

Additional Editor Comments:

One of the review recommend the rejection on your paper. Please think carefully about these comments and wrote your rebuttal letter.

Reviewers' comments:

Reviewer's Responses to Questions

**Comments to the Author**

1. Is the manuscript technically sound, and do the data support the conclusions?

Reviewer #1: No

Reviewer #2: Partly

Reviewer #3: Yes

2. Has the statistical analysis been performed appropriately and rigorously? 

Reviewer #1: Yes

Reviewer #2: No

Reviewer #3: Yes

3. Have the authors made all data underlying the findings in their manuscript fully available?

Reviewer #1: Yes

Reviewer #2: Yes

Reviewer #3: No

4. Is the manuscript presented in an intelligible fashion and written in standard English?

Reviewer #1: Yes

Reviewer #2: Yes

Reviewer #3: Yes

5. Review Comments to the Author

Reviewer #1: The intention of this work can be very useful, but from my point of view there was a lack of better planning in the experiments, a better presentation of the results and a better biological explanation of what was found. More specific observations and suggestions can be found in the attached file.

Reviewer #2: GENERAL COMMENTS

It is an interesting manuscript that addresses a general and widespread problem in beekeeping. Many research groups are trying to evaluate different methods for Varroa control. Especially, organic compounds that are compatible with the activity are investigated. The use of essential oils or organic acids is not original per se but there is still room to investigate.

The term "multi-method study" is a bit bombastic to say that a study is performed to analyse the effectiveness of a treatment. Whenever new treatments are evaluated, their effect is analyzed using different techniques, approaches or analyzing different responses, that does not make it special.

SPECIFIC COMMENTS

Line 31: Four treatments were proved because the control group is a treatment level too.

Line 33: "juice cocktail" composed by?

Line 40-41: It is not exceptional, it is the correct method to conduct a trial to prove the effectiveness of a treatment.

Line 94: I´m not sure if they are new compounds.

Line 102: It is not true, multi-methods approach is a common approach.

Line 112: There were five treatments groups, the control group is another treatment lever.

Figure 2: It is not necessary.

Lines 164-166: The composition of this fruit juice is not very clear. Perhaps a chemical analysis could help elucidate its composition. More information is needed.

Line 304: As far as I can understand, the same beehives were sampled at different times of the trial to perform certain analyses. It is clear that this is a repeated measures design and should be evaluated as such. Calculating the absolute change of the variables to be studied does not seem to make sense to me. I cannot conclude that the entire statistical analysis is wrong because, in essence, it may not be, but I believe that the study would greatly benefit from the application of repeated measures ANOVA or generalized linear models.

Line 338-340: This sentence is repeated.

Line 343: "appeared to be non-toxic to bees". Did it prove objectively?

Line 443: This effect was a consequence of the treatment? One day post-treatment?

Line 489-495: Repeated

Line 504-504: What was the expected effect size? What power was the proposed experimental design?

Line 570-571: This conclusion was not supported by the results.

Line 609-610: this was not supported by the results. This effect was not significant. Also, there was no pattern in the vitellogenin expression.

Reviewer #3: This work provides a multi-method approach to investigate the effects of two essential oils (oregano and cinnamon) and a mixed fruit juice on the health and productivity of honey bee colonies. The study and development of natural-based acaricide treatments are important to avoid the emergence of resistance by Varroa mites. The novelty of this work consists in the evaluation of multiple parameters of colony health and productivity (i.e., colony size, honey production, Varroa infestation level, pathogens infection, and expression levels of immune-related genes) following the administration of sugar syrup supplemented with the tested natural compounds.

I generally find the work of interest to the apiculture sector and suitable for publication. However, I find some revisions necessary.

INTRODUCTION

• Lines 64-81: I think there are very few references in this paragraph, are there others available in the literature to be added? For instance, what are the main discrepancies found between field studies (line 69)? Can the authors provide references? Same for lines 74 and 79.

• Line 81: “eliminate or reduce diseases in honey bee populations”. In my opinion, the term 'eliminate' is too strong. The pathogens considered in this work, and in general pests and pathogens of honey bees, can never be totally eliminated. The strategy is to implement the best practices to ensure good bee health and welfare and a good nutritional level so that the bees are able to coexist in balance with these stressors.

• Lines 82-86: is there only one reference for this statement? Please add more.

• Line 84: I do not know whether the term “preferring” is appropriate, since, as the authors themselves write below, this “preferring” only occurs under particular conditions. Perhaps it would be better to remove this term and just say that bees have been observed foraging fruit juice in certain conditions.

• Line 86: “The dietary supplementations…”. Better link to the previous sentence.

• I think the authors have to motivate the choice of specifically these two essential oils (oregano and cinnamon).

MATERIALS AND METHODS

• The abbreviations used to indicate the treatments are a part in English (ORE for oregano) and a part in Italian (CAN for “cannella”, the Italian term for cinnamon). Please put everything in English.

• Is there a reason for the choice of these times of treatment, bee collection, and measurement of the level of varroa infestation?

• Line 153: why was the 1:1 ratio of syrup:supplement chosen? What are the actual administered concentrations of essential oils and fruit juice?

• Line 162: could soya lecithin have had any effect on bees? Is there any information in the literature?

• Line 164: in my opinion, the nutritional values of fruit-cocktail are missing. Since this supplement is given to bees for its content in vitamins, minerals, fiber and other phytochemicals, I think it would be appropriate to include the concentration of these elements.

• Line 173: how much Api Bioxal was applied? In which concentration?

• Line 179: the two different methods were presumably used for two different purposes. That is, the powdered sugar only to measure the difference in infestation between beginning and end, and the mites fall to measure the effectiveness of each treatment at each time? Specify this in this section.

• Lines 212-216: please motivate the choice of these pathogens.

• Lines 296-301: why was honey production not measured by weighing? Weighing the extracted honey, or the difference in weight between full and empty honey suppers could give more readily understandable quantitative data that could also possibly be cited by other papers.

• Were non-parametric tests used for statistical analysis because the data were not normally distributed? Please, include in section 2.8 the test performed to assess the distribution of the data.

RESULTS

• Lines 338-340: maybe it is unnecessary to repeat it.

• Line 403: “the fungal N. ceranae”. Nosema is no longer classified as a fungus, please correct this, also in line 542.

• Lines 463-564: this information is actually not visible in the graph. The boxes represent all the four colonies of each treatment, the single hive 13 cannot be shown.

• Line 466: a new paragraph on honey production begins from this line. Please enter the corresponding subtitle.

DISCUSSION

• Line 495: I would better explain the initial intention. It cannot be "better describe the observed differences" since in the experimental design phase one cannot know whether differences will actually be observed.

• Lines 496-500: In my opinion, this part is better moved to the end of the introduction, together with the aim of the work.

• Line 501: “bee survival”. There is no evidence of measurement of this parameter in the text. Are the authors referring to colony size? But they are two different things.

• Lines 516-557: the part of the discussion on pathogens seems to me to be a little off-topic. The aim of the work is not an epidemiological study, but an investigation of the effects of treatments on colonies. This part of the discussion should focus more on that. It seems to me that it digresses a little too much on epidemiological aspects and that the results are repeated too much. The last sentence, 554-557, is the gist of the findings.

• Lines 526-527: actually, CBPV is often found in asymptomatic colonies.

• Line 561: “with the exception of the hive with infected brood by M. plutonius”, but is this statistically significant?

• Lines 570-571: this is a conclusion drawn from the literature, not from the present work.

• Lines 582-584: anything in the literature about this?

FIGURES

• Figure 1 caption: here the times of supplement feeding are T0, T10, and T11. Later in the text, it is written that the times are T0, T10, and T24. Clarify and if necessary, correct these times. Furthermore, more than the initially declared 10 days elapsed between the administration at T10 and the administration at T24.

• Figure 3: such long captions are difficult to read to me. It could be shortened by removing details already explained in the text of the paper. For example, lines 352-354 and lines 357-360 could be deleted. Same comment for figures 4, 5, 7.

I would add “Number of mites” in the Y-axis title and “treatments” in the X-axis title.

• Figure S1: I would add “Number of mites” in the Y-axis title and “treatments” in the X-axis title.

• Figure 4: In panel B, the names of the treatments on the X-axis are missing. In panel C, add a parenthesis or a connecting line between the boxes that are significantly different, for better clarity of visualization. Does the single asterisk indicate p<0.05? Include it in the caption.

• Figure 5: state in the caption what the dotted line represents. Uniform T55 or T1M throughout. I would add “Difference in growth” in the Y-axis title and “treatments” in the X-axis title.

• Figure 6 seems unnecessary to me, consider whether eliminate it.

6. PLOS authors have the option to publish the peer review history of their article (what does this mean?). If published, this will include your full peer review and any attached files.

Reviewer #1: No

Reviewer #2: No

Reviewer #3: No

---

## [Author Response · Author response to Decision Letter 0]

1 Mar 2024

Reply to Reviewer 1

Q1 (line 1): This title does not reflect the content of the article, which is based on evaluating the effect of natural products on the varroa mite (viruses, bacteria, fungi, vitellogenin, etc), colony growth and honey production.

Q2 (lines 94-96): The objective does not coincide with the title of the article, which should be written in this same sense.

R1-2: We agree with Reviewer 1 and we changed the title, as suggested:

Effects of natural treatments on the varroa mite infestation levels and overall health of honey bee (Apis mellifera) colonies

Q3 (line 112): Did these hives started under the same bee population, bee frames and weight conditions? They must have homogenized them before starting the treatments.

 R3: The hives were from the same apiary and were divided into five homogenous groups. It was clarified in the text, lines 112-114.

Q4 (line 113): From my point of view this is the biggest problem in this article. I consider that only three applications of the treatments is not enough to see the effect on the variables that were recorded. I consider that due to this situation no significant differences were observed in the results. The bees need to be consuming the products that are supplied to them for a longer period of time to be able to verify if there is an effect. 

Additionally (and the same authors mention it) the number of colonies used is low, which could not have allowed obtaining more conclusive results.

R4: We understand the reviewer’s concern. The dose and length of EO-based treatments were also our concern in planning the experiments. Since no standard protocols or guidelines for the use of essential oils (EOs) in apiary exist, we reviewed the literature in order to answer some of the questions about the use of EOs in field conditions: “how to treat” the bees (e.g. vaporization, spraying, exposure to EO-embedded strips or EO-supplemented diets), “how much to use” and “how long to use” in order to reduce the mite infestation levels with low impact on bee colony health. Most recent literature on the topic investigated the toxicity of EOs toward the bees using fumigation as method of administration (Aglagane et al., 2021; Bava et al., 2023). Limited researches focused on the effects of the oral administration of EOs to bees. Please, see below some examples.

Glavan et al., (Pesticide Biochemistry and Physiology, 2020) evaluated the sublethal effects of the oral exposure of carvacrol and thymol for 7 days in honeybee workers. The authors tested a wide range of concentrations for carvacrol (0.005, 0.05, 0.5 and 5%) and thymol (0.05, 0.5 and 1%). The authors found that 0.05% or higher carvacrol or thymol exposure oral concentrations affected honey bee nervous system. They concluded that prolonged treatments with thymol and carvacrol may have sublethal effects on bees. 

Rossini et al (PLOS ONE, 2020) studied the long-term effects of the oral administration of Epatorium buniifolium EO-supplemented syrups given to honey nurses for 12 consecutive days under laboratory conditions. EOs were added to sugar syrup to reach final concentrations of 300 (0.03%), 3000 (0.3%) and 6000 (0.6%) ppm. While food consumption and survival were not affected by the EO treatments, the cuticular hydrocarbons, which mediate social recognition in honeybees, changed in composition. The authors concluded that the ingestion of EOs can affect the composition of cuticular hydrocarbons and that the effect is dose-dependent, emphasizing the importance to address the long-term effects of this type of products on honeybee performance before their use in beekeeping.

Gende and colleagues (Bulletin of Insectology, 2009) tested in an apiary trial the efficacy of two doses of a syrup supplemented with the cinnamon EO at 1000 µg/mL (0.1%). The syrup was given to the bees at a 7-day interval and tested against artificially infected Paenibacillus larvae colonies. The EO-based treatment reduced the incidence of infected larvae and showed low toxicity effects to bees. 

Canchè-Colli and colleagues (PeerJ, 2021) administered three doses of EO-supplemented diets to bees at a 3-day interval. The EOs were added to the base formulation at 1% proportion. Although the survival probability curves of the treated bees were lower than in the unsupplemented control group, no statistical differences were found.

On the basis of all the above findings, namely the effects of EOs on honey bee survival, nervous system and social recognition, we decided to administer three doses of EO-supplemented syrups - 500 µL of EO in 1 L syrup (0.05%) - at intervals of about 10 days. 

Concerning the sample size, we also evaluated the literature. Canchè-Colli et al. (2021) used three cages (24-30 bees inside individual cages) per group to assess the effects of EO-enriched diets. They found significant differences in immune-related gene expression of Vg, proPO and GOx. Porrini et al. (2017) used three replicates of 25-35 bees per each treatment. They found that the oral administration of EOs significantly affected the Apis mellifera survival. Ewert et al., (2023) treated three cages (about 50 adult bees each) per group with EO-supplemented syrup, as well. The authors described the effects on bee life span, gene expression and gut microbiota abundance. Gende et al. (2009) used five hives per group to assess the effect of a cinnamon-enriched syrup. The found that the cinnamon EO reduced the P. larvae infection in apiary.

In our study, four hives per group were sufficient to reveal significant differences on immune gene expression between treatment and control groups. On the other hand, we did not find effects against V. destructor. The reason may be due in part to the small sample size, which reduces the power to detect significance. Other factors may have contributed to the inefficacy of the treatments, such as an inappropriate dosage formulation (500 μL/L), unsuitable application method (oral administration) of EOs used or both of them. Therefore, further studies are need to shed light on the potential uses of EOs in beekeeping.

Q5 (line 121): The image of an apiary does not have much relevance for it to appear in an article! 

R5: We removed the image of the apiary, as suggested.

Q6 (line 125): Instead of using four colonies per treatment, you could have used 10

R6: We discussed above the choice of sample size. Please, see R4.

Q7 (line 126): This description is repeated many times throughout the document. It is enough to make a good description the first time and then just write down an abbreviation of the treatments.

R7: We eliminates redundant sentences from the text, as suggested. 

Q8 (line 292): For this variable, it is not enough to count the number of frames with bees; to estimate the growth of the colonies, the number of cells with brood must be counted and their weight increase periodically recorded.

R8: We agree with the reviewer, the method we used did not estimate the colony growth but the adult bee population density, as per Chaimanee et al., 2021. We cited this study in Materials and Methods. We therefore replaced “colony growth” with “adult bee population density” or “colony density”.

Q9 (line 296): Also is not enough to count the number of frame with honey. You must have weighed the supers with honey, extracted it and then weighed them again and thus obtained the honey production of each colony in kilograms.

R9: We agree with the reviewer that, for a better accuracy, we should have extracted honey from the frames and weighed it, as suggested. However, we measured the honey yield of five hives located in the same apiary and left untreated. These five hives were excluded from the experiment and used by the beekeeper for his own honey production. We found that a full 9-frame honey supper held around 15 kgs of honey, while a frame held around 1.5 kgs. We used these data to estimat the amount of honey produced by our colonies and assigned a score, as described in Materials and Methods. As we believe that the scoring system may be confusing, we replaced it with estimated weights (kg). We reperformed the statistical analysis and the graph. No significant differences between treatment and control groups were confirmed. We clarified that better in Materials and Methods (paragraph 2.7). 

Q10 (line 338): The objectives of the article do not consider verifying whether the treatments are toxic or not toxic to the bees, therefore, what you wrote here does not make sense.

R10: The issue of EO toxicity towards the adult bees has been investigated by exposure bioassays in the literature. Limited researches have evaluated the EO toxicity in the EO-supplemented diet for bees. We therefore believe that, although the toxicity of the treatments was not an objective of the study, the safety of the tested natural treatments we have observed under our experimental conditions, deserves to be mentioned. 

Q11 (line 346): This information has already been written in the materials and methods. You don't have to repeat it again. In the results you have to be concrete, specific and only write down what was found.

 R11: We removed the redundant sentence, as suggested.

Q12 (line 364): I am only going to comment in this paragraph on the fact that you did not find statistical differences, which could have been due to two situations: 1.- the short time that the bees were exposed to the treatments and 2.- the low number of colonies that were used. 

R12: We do not know what makes the tested natural treatments ineffective against V. destructor. It is likely that the low number of colonies per group or the short time of the treatments may have affected the results, as suggested by the reviewer. It is equally likely that other factors may have contributed to the inefficacy of the treatments, namely the dose or the application method here used, or both of them, as discussed above (Please, see R4).

 Q13 (line 386): These results may be interesting, but no trend or direct relationship is observed with the treatments used in the discussion.

R13: We agree with the reviewer, the results of the molecular diagnosis of DNA and RNA pathogens are interesting. However, we could not estimate the effects of the treatments against those pathogens because of the limited number of cases here described or the lack of differences between the control and the treatment groups, as discussed in the manuscript (Please, see Discussion, lines 544-547). 

Q14 (line 451): Here again what was already specified in the materials and methods is repeated.

Q15 (line 454): This should not be noted down again, just seeing the figure should be enough.

R14-15: The lines in question refer to the legend of Fig. 5. We referred to the PLOS ONE guidelines for authors for writing the Figure legends.

 Q16 (line 474): Likewise, an image of the apiary with the honey supers does not have much value in a publication, what has more value is a table with quantitative data, in this case it should have been the average number of kilograms of honey per treatment with its respective deviation standard.

R16: We removed the image of the apiary with the honey supers, as suggested, and renumbered the figures accordingly. 

Q17 (line 489): Again, this paragraph is repetitive information with previous information that has already been written.

R17: As suggested, we shortened the first paragraph of Discussion. 

Q18 (line 504): Here you yourselves refer to the limitation of your study for the first time...

R18: We addressed this limitation also later in the text (see lines 572-573).

Q19 (line 513): This may be confirmation that the treatments had no effect on the varroas due to the short time you applied

 R19: This issue has been addressed above. Please, see R4 and R12.

Q20 (line 516): Ok, this is good information, but how do you relate it to the treatments used in this work? This same observation applies to almost the entire discussion.

R20: This issue has been discussed above. Please, see R13.

Q21 (page 27): This may be confirmation that the treatments had no effect on the varroas due to the short time you applied

 R21: This issue has been discussed above. Please, see R4 and R12.

Q22 (line 581): What evidence is based on to support this statement?

R22: We removed the sentence from Discussion, as no evidence was found to support that statement. 

Q23 (line 589): Here you mention for the second time the limitation due to having used a low number of colonies.

R23: It is likely that both the small group size and the high intra-group variability might have influenced our research outcomes, as mentioned in Discussion.

Q24 (Figure 3): This is not the best way to express the results. The figures must be self-understanding, they must have subtitles on the horizontal and vertical axes. It is better to write down quantitative data in a table (means and standard deviations). This same observation is for all the other figures.

R24: All figures have subtitles on horizontal and vertical axes. Moreover, detailed figure legends are provided to allow readers to understand them without referring to the text. We believe that charts provide a quicker comprehension of the data by allowing the identification of patterns, trends and relationships. We are, however, available to add the corresponding tables as supplemented materials, if requested.

Reply to Reviewer 2

Q1: It is an interesting manuscript that addresses a general and widespread problem in beekeeping. Many research groups are trying to evaluate different methods for Varroa control. Especially, organic compounds that are compatible with the activity are investigated. The use of essential oils or organic acids is not original per se but there is still room to investigate.

The term "multi-method study" is a bit bombastic to say that a study is performed to analyse the effectiveness of a treatment. Whenever new treatments are evaluated, their effect is analyzed using different techniques, approaches or analyzing different responses, that does not make it special.

R1: We agree with the reviewer and removed the term “multi-method study" from the title (a new title has been proposed), Abstract, Introduction and Discussion.

SPECIFIC COMMENTS

Q2, Line 31: Four treatments were proved because the control group is a treatment level too.

 R2: We clarified better in Abstract as follows:

The colonies were divided into five groups: four treatment groups and one control group.

Q3, Line 33: "juice cocktail" composed by?

R3: We replaced the term “juice cocktail” with “mixed fruit cocktail juice” in Abstract. Details on its composition are provided in Materials &Methods (paragraph 2.2) 

Q4, Line 40-41: It is not exceptional, it is the correct method to conduct a trial to prove the effectiveness of a treatment.

R4: The sentence was removed from Abstract.

Q5, Line 94: I´m not sure if they are new compounds.

R5: We removed the term “new” from the sentence.

Q6, Line 102: It is not true, multi-methods approach is a common approach.

R6: We removed the sentence from Introduction.

Q7, Line 112: There were five treatments groups, the control group is another treatment lever.

R7: It was better clarified in the legend of Fig.1

Q8, Figure 2: It is not necessary.

R8: We removed Fig. 2, as suggested

Q9, Lines 164-166: The composition of this fruit juice is not very clear. Perhaps a chemical analysis could help elucidate its composition. More information is needed.

R9: The composition of the fruit juice was provided, as per the manufacturer.

Q10, Line 304: As far as I can understand, the same beehives were sampled at different times of the trial to perform certain analyses. It is clear that this is a repeated measures design and should be evaluated as such. Calculating the absolute change of the variables to be studied does not seem to make sense to me. I cannot conclude that the entire statistical analysis is wrong because, in essence, it may not be, but I believe that the study would greatly benefit from the application of repeated measures ANOVA or generalized linear models.

R10: We agree with the reviewer that beehiv

---

## [Decision Letter · Decision Letter 1]

12 Mar 2024

PONE-D-23-42744R1A multi-method field study to investigate the effects of natural treatments on the health of honey bee (Apis mellifera) coloniesPLOS ONE

Dear Dr. Marianelli,

Thank you for submitting your manuscript to PLOS ONE. After careful consideration, we feel that it has merit but does not fully meet PLOS ONE’s publication criteria as it currently stands. Therefore, we invite you to submit a revised version of the manuscript that addresses the points raised during the review process.

We look forward to receiving your revised manuscript.

Kind regards,

Kai Wang

Academic Editor

PLOS ONE

Reviewers' comments:

Reviewer's Responses to Questions

**Comments to the Author**

1. If the authors have adequately addressed your comments raised in a previous round of review and you feel that this manuscript is now acceptable for publication, you may indicate that here to bypass the “Comments to the Author” section, enter your conflict of interest statement in the “Confidential to Editor” section, and submit your "Accept" recommendation.

Reviewer #2: All comments have been addressed

Reviewer #3: (No Response)

2. Is the manuscript technically sound, and do the data support the conclusions?

Reviewer #2: Yes

Reviewer #3: Partly

3. Has the statistical analysis been performed appropriately and rigorously? 

Reviewer #2: No

Reviewer #3: Yes

4. Have the authors made all data underlying the findings in their manuscript fully available?

Reviewer #2: Yes

Reviewer #3: No

5. Is the manuscript presented in an intelligible fashion and written in standard English?

Reviewer #2: Yes

Reviewer #3: Yes

6. Review Comments to the Author

Reviewer #2: Line 115: The experimental design had a treatment that has five levels, the control group is one more level of the treatment. This may seem like a minor point and does not affect the coherence of the study, but it does address the use of terminology specific to the experimental design.

The authors state that they do not know the frequency distribution of the response variables. This is obvious before carrying out the study, then with the data obtained they evaluated whether they met the assumption of normality. Not being proven, they decided, correctly, to use non-parametric methods. I believe that the statistical approach used by the authors is not entirely correct, since, with the frequency distribution they can evaluate which distribution they can fit, choose the appropriate link function, and use a Generalized Linear Model, which is a method much more appropriate.

On the other hand, it is possible to incorporate into generalized linear models the random effect of having measured the same hive on several occasions throughout the trial. I suggest that the authors, evaluate their data using a Generalized Linear Model of repeated measures, using the distribution that best fits the observed data, and select the appropriate link function. I consider that the statistical method used by the authors is not wrong, it is just not the most appropriate.

I understand the authors' concern about the size of each treatment level, undoubtedly one of the most relevant limitations of the study. However, once the limitation was recognized, what I requested in the previous review was that they support the sample size used. I understand the authors about they did not have an expected effect size before the trial and they used similar approaches from other works. This is not wrong in itself, but it does not fully clarify the point. When you design an experiment, you establish the sample size and if you do not have an idea of the expected effect, you can make a priori hypotheses. However, if the sample size was decided based on previous studies, what I requested is that you estimate the power of the design used, which can provide evidence of the limitation. The power calculation will tell you the probability that they made a Type II error.

Reviewer #3: The authors responded to the comments and revised much of what was requested. I still have some comments, which had not been fully resolved previously.

- Line 165: the percentages of the different fruits that make up the juice have been included, but in my opinion the analytical composition (e.g. vitamins, sugars, minerals, fibres...) is still missing in order to know what was actually fed to the bees.

- Line 174: the concentration of oxalic acid administered with the treatment is still missing. It is indicated on the product label.

- Line 215: the authors stated that they chose the pathogens most prevalent in the apiary. Do they refer to previous pathological analyses? Do they refer to literature data?

- Line 348: I still do not understand the 'background mortality'. It is mentioned here for the first time, but how is it measured? Where is the explanation of its measurement and statistical analysis in the materials and methods?

- Line 489-490: why does the reader have to get to the bottom of the article to know the reason for the choice of essential oils?

- I am still unsure about the method of estimating the amount of honey produced. There was no need to extract the honey, it would have been sufficient to weigh the empty supers and reweigh them again once they were filled with honey.

- Paray et al. (doi.org/10.1016/j.sjbs.2020.11.053) reviewed honey bee nutrition and pollen sobstitutes. There is also a reference to a work where different fruit juices were fed to bees. Perhaps this could be useful for introduction and/or discussion.

7. PLOS authors have the option to publish the peer review history of their article (what does this mean?). If published, this will include your full peer review and any attached files.

Reviewer #2: No

Reviewer #3: No

---

## [Author Response · Author response to Decision Letter 1]

25 Mar 2024

Reply to Reviewer 2

Q1: Line 115: The experimental design had a treatment that has five levels, the control group is one more level of the treatment. This may seem like a minor point and does not affect the coherence of the study, but it does address the use of terminology specific to the experimental design.

R: We have taken note of the clarification and have amended the text accordingly.

Q2: The authors state that they do not know the frequency distribution of the response variables. This is obvious before carrying out the study, then with the data obtained they evaluated whether they met the assumption of normality. Not being proven, they decided, correctly, to use non-parametric methods. I believe that the statistical approach used by the authors is not entirely correct, since, with the frequency distribution they can evaluate which distribution they can fit, choose the appropriate link function, and use a Generalized Linear Model, which is a method much more appropriate.

On the other hand, it is possible to incorporate into generalized linear models the random effect of having measured the same hive on several occasions throughout the trial. I suggest that the authors, evaluate their data using a Generalized Linear Model of repeated measures, using the distribution that best fits the observed data, and select the appropriate link function. I consider that the statistical method used by the authors is not wrong, it is just not the most appropriate.

R: We agree that GLM would be the most appropriate method to analyse this type of data, provided that the distribution of the data is known or can be determined with confidence. In our experiment, the number of hives per group was rather small (n=4), and we did not feel confident to base the determination of the data distribution and, consequently, of the link function, whose choice affects the reliability of the GLM results, on such a limited number of hives. For this reason, we preferred to use the non-parametric method, which is reliable regardless of the exact identification of the data distribution.

Q3. I understand the authors' concern about the size of each treatment level, undoubtedly one of the most relevant limitations of the study. However, once the limitation was recognized, what I requested in the previous review was that they support the sample size used. I understand the authors about they did not have an expected effect size before the trial and they used similar approaches from other works. This is not wrong in itself, but it does not fully clarify the point. When you design an experiment, you establish the sample size and if you do not have an idea of the expected effect, you can make a priori hypotheses. However, if the sample size was decided based on previous studies, what I requested is that you estimate the power of the design used, which can provide evidence of the limitation. The power calculation will tell you the probability that they made a Type II error.

R: The sample size per group used in our study (n=4) would have allowed us to detect an effect of very large magnitude (Cohen's d = 3.2) with multiple comparisons performed by the Mann-Whitney U test (min ARE distribution), with one-tailed alpha = 0.0125 (i.e., experimentwise alpha = 0.05 using Bonferroni's correction for 4 multiple comparisons) and power = 0.80. Under the same conditions of alpha, tails and statistical test, the power to detect an effect of medium to large size (Cohen's d = 1) would have been extremely low (1-beta = 0.11). In any case, the differences in mite infestation levels between the treatment and control groups were very small and mostly in the opposite direction to what we expected, suggesting a lack of efficacy of our treatments. This finding suggests that the lack of significance in the differences is due to a real lack of effect rather than a problem of low power. To give you a complete picture, we would have needed 24 hives per group (120 hives in total) to detect a Cohen's d effect size of 1.0, using the same statistical parameters (one-tailed alpha = 0.0125 and power = 0.80), making it impossible to carry out the desired experiment.

Reply to Reviewer 3

Q: The authors responded to the comments and revised much of what was requested. I still have some comments, which had not been fully resolved previously.

- Line 165: the percentages of the different fruits that make up the juice have been included, but in my opinion the analytical composition (e.g. vitamins, sugars, minerals, fibres...) is still missing in order to know what was actually fed to the bees.

R1: We added the nutritional information of Cuore di Frutta, as it appears on the label of the juice. Please, see lines 166-166. There is no nutritional information on the blueberry juice.

- Line 174: the concentration of oxalic acid administered with the treatment is still missing. It is indicated on the product label.

R2: We included the oxalic acid concentration. 

- Line 215: the authors stated that they chose the pathogens most prevalent in the apiary. Do they refer to previous pathological analyses? Do they refer to literature data?

R3: The literature was cited and we replaced the term 'the most prevalent' with 'common'. 

- Line 348: I still do not understand the 'background mortality'. It is mentioned here for the first time, but how is it measured? Where is the explanation of its measurement and statistical analysis in the materials and methods?

R4: The number of dead bees inside each hive was recorded 24 hours after the first treatment. No significant differences were found compared to the control group, which experienced background mortality. However, measurements were not repeated after the second and third treatments. We did not mention this in the Materials and Methods section as it was outside the scope of our research. As we did not thoroughly investigate the toxicity of our treatments towards bees, we removed any references to the safety of the tested treatments from the abstract, methods, and discussion.

.

- Line 489-490: why does the reader have to get to the bottom of the article to know the reason for the choice of essential oils?

R5: We agree with the reviewer and relocated the sentence in the Introduction section. Please, see lines 94-96.

- I am still unsure about the method of estimating the amount of honey produced. There was no need to extract the honey, it would have been sufficient to weigh the empty supers and reweigh them again once they were filled with honey.

R6: We agree with the reviewer that our estimation of honey production was inaccurate. Unfortunately, we are unable to rectify this. However, if the reviewer believes that our method is fundamentally flawed, we are willing to remove any references to honey production from the text. 

- Paray et al. (doi.org/10.1016/j.sjbs.2020.11.053) reviewed honey bee nutrition and pollen sobstitutes. There is also a reference to a work where different fruit juices were fed to bees. Perhaps this could be useful for introduction and/or discussion.

R7: We greatly appreciate the reviewer's suggestion. In our introduction, we referenced the study conducted by Pande et al. (2015), which examined the use of four different fruit juices as supplements for bee feeding. Please, see lines 85-87 and 92-93.

---

## [Decision Letter · Decision Letter 2]

10 Apr 2024

Effects of natural treatments on the Varroa mite infestation levels and overall health of honey bee (Apis mellifera) colonies.

PONE-D-23-42744R2

Dear Dr. Marianelli,

We’re pleased to inform you that your manuscript has been judged scientifically suitable for publication and will be formally accepted for publication once it meets all outstanding technical requirements.

Kind regards,

Kai Wang

Academic Editor

PLOS ONE

Additional Editor Comments (optional):

Reviewers' comments:

Reviewer's Responses to Questions

**Comments to the Author**

1. If the authors have adequately addressed your comments raised in a previous round of review and you feel that this manuscript is now acceptable for publication, you may indicate that here to bypass the “Comments to the Author” section, enter your conflict of interest statement in the “Confidential to Editor” section, and submit your "Accept" recommendation.

Reviewer #3: All comments have been addressed

2. Is the manuscript technically sound, and do the data support the conclusions?

Reviewer #3: Yes

3. Has the statistical analysis been performed appropriately and rigorously? 

Reviewer #3: Yes

4. Have the authors made all data underlying the findings in their manuscript fully available?

Reviewer #3: Yes

5. Is the manuscript presented in an intelligible fashion and written in standard English?

Reviewer #3: Yes

6. Review Comments to the Author

Reviewer #3: The authors have responded to the reviewers' comments and have improved the manuscript as requested. I consider the manuscript in this form to be suitable for publication.

7. PLOS authors have the option to publish the peer review history of their article (what does this mean?). If published, this will include your full peer review and any attached files.

Reviewer #3: No
